# GraphEx: A User-Centric Model-Level Explainer for Graph Neural Networks

**Sayan Saha**
Indian Statistical Institute, Kolkata

**Monidipa Das**
IIT (ISM), Dhanbad

**Sanghamitra Bandyopadhyay**
Indian Statistical Institute, Kolkata

## Abstract

With the increasing application of Graph Neural Networks (GNNs) in real-world domains, there is a growing need to understand the decision-making process of these models. To address this, we propose GraphEx, a model-level explainer that learns a graph generative model to approximate the distribution of graphs classified into a target class by the GNN model. Unlike existing methods, GraphEx does not require another black box deep model to explain the GNN and can generate a diverse set of explanation graphs with different node and edge features in one shot. Moreover, GraphEx does not need white box access to the GNN model, making it more accessible to end-users. Experiments on both synthetic and real datasets demonstrate that GraphEx can consistently produce explanations aligned with the class identity and can also identify potential limitations of the GNN model.

## 1 Introduction

The deployment of Graph Neural Networks (GNNs) in applications such as drug discovery has necessitated that they become explainable to the end users. Current research on GNN explainability can be broadly classified into two categories: instance-level explanations and model-level explanations. Instance-level explanations aim to provide explanations specific to the input data, while model-level explanations aim to provide a high-level understanding of the model's decision-making process, shedding light on its general behavior. Model-level explanations require less human supervision, can be generated efficiently, and offer valuable insights into when the model can be trusted. Despite the significant advantages of model-level explanation methods for GNNs, they have received less attention in research compared to instance-level explanation methods. We propose a novel generative model for graphs based on likelihood maximization that can serve as a model-level explainer. Our model can create graphs of varying sizes, incorporating different node and edge features in a single step. In comparison to existing literature, our method presents several significant advantages. Unlike XGNN(Yuan et al., 2020), our approach does not rely on another black box deep learning model to explain GNNs. Furthermore, unlike GNNInterpreter(Wang & Shen, 2022), our method does not require white box access to the GNN model being explained, which is crucial for making the explainability scheme accessible to end-users who typically do not have access to the model.

## 2 GraphEx

The objective of a model-level explainer is to provide insight into the decision-making process of the model by producing instances that elicit a specific response from the model. This includes generating typical examples that the model confidently classifies into a target category and examples that lie on the decision boundary which is crucial for understanding how the model distinguishes between different classes. Aligned with this goal, GraphEx uses a generative model trained on graphs classified into a target class by the GNN classifier to construct representative examples belonging to the target class or the decision boundary. Next, we describe our generative model.

We formulate a graph representation model for representing any graph with atmost $N$ nodes having $k$ node types and $e$ edge types. Such a graph can be represented using a tuple $(X, E)$ where $X \in \mathbb{R}^{N \times (1+k)}$ is the node feature matrix and $E \in \mathbb{R}^{N \times N \times (1+e)}$ is the edge type matrix. The node types range from 1 to $k$ and the edge types range from 1 to $e$. For each node $i$, if $X(i, 0) = 1$, then the node $i$ is absent, otherwise if $X(i, a) = 1$ for $a \in \{1, .., k\}$, then node $i$ is of type $a$. Similarly,

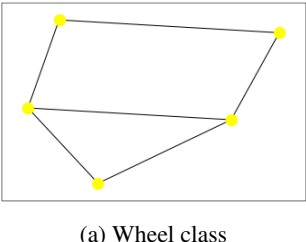

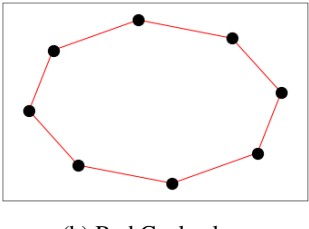

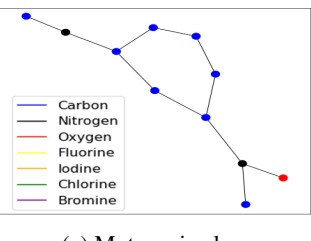

(a) Wheel class               (b) Red Cycle class             (c) Mutagenic class

Figure 1: Explanations for different target classes

if $E(i, j, 0) = 1$, then the corresponding edge between the $i^{th}$ and the $j^{th}$ node does not exist. Otherwise, if $E(i, j, a) = 1$ where $a \in \{1, ..., e\}$ then the corresponding edge is of type $a$. We use the graph representation model to generate explanation graphs by considering each node $X_i$ and edge $E_{ij}$ as a random variable that follows a Categorical distribution. Formally,

$$\begin{cases} x_i \sim Categorical(\boldsymbol{\theta}_i) & \boldsymbol{\theta}_i = \{\theta_{i0}, ......, \theta_{ik}\} \\ e_{ij} \sim Categorical(\boldsymbol{\eta}_{ij}) & \boldsymbol{\eta}_{ij} = \{\eta_{ij0}, ....., \eta_{ije}\} \end{cases} \quad (1)$$

Assume, that we are given a dataset of graphs $D$ in which a graph can belong to a class $c$, where $c \in \{1, ...., C\}$. We are also given a trained graph classifier $f(.)$, which has learnt to predict the class of a graph from this dataset. Treating a graph as a random variable $G$, we assume that,

$$G|f(G) = c \sim \prod_{i=1}^{N} Categorical(\boldsymbol{\theta_{i,c}}) \prod_{j,k=1}^{N} Categorical(\boldsymbol{\eta_{jk,c}}) \quad (2)$$

Hence, given a set of graphs $D_c$ which $f(.)$ has labelled as belonging to a target class $c$ we estimate parameters as shown in Appendix A for this generative model such that it can generate graphs which belong to a distribution close to the distribution of $D_c$ and which $f(.)$ would label as $c$.

## 3 EXPERIMENTAL STUDY

We evaluate our method on both synthetic and real datasets. We train a GNN graph classifier on each dataset and then employ GraphExto explain the classifier.We created two distinct graph datasets, the **Wheel-Tree** and the **ColoredCycles**. The **Wheel-Tree** dataset consists of two classes of graphs, with one class being the **Wheel** graph and the other class being the Tree graph. A **Wheel** graph comprises of a hub node connected to a cycle of **n-1** nodes whereas a Tree graph **Tree** graph,is a full r-ary tree in this context. The **ColoredCycles** dataset includes cycle graphs with categorical edge features. The two classes of graphs in this dataset are defined by their edge color, with one class having green cycles and the other class having red cycles.We also evaluate GraphEx on the **MUTAG**(Debnath et al., 1991) dataset which consists of mutagenic and non-mutagenic classes of molecules. We describe the dataset generation and the GNN classifiers in Appendix B

GraphEx achieved a comprehensive understanding of the model's behavior on the **Wheel-Tree** dataset. We consistently observe high class confidence for the Wheel class in graphs where all nodes form a cycle, and for the Tree class in graphs without any cycles. The decision boundary of the classifier is determined by graphs that contain a mix of nodes belonging to cycles and those that do not. On **ColoredCycles** dataset, GraphEx produces class-consistent explanations with high confidence scores for both classes of graphs. Specifically, we obtain red cyclic graphs to explain the class with red cycles and green cyclic graphs to explain the class with green cycles.GraphEx can detect distinctive patterns that influence the classifier's decision in assigning class labels to molecules for the **MUTAG** dataset. For the mutagenic class, the presence of $N$ and $O$ atoms, raises the likelihood of the molecule belonging to the mutagenic class. On the other hand, the high-scoring explanations for the non-mutagenic class predominantly feature only $C$ atoms indicating a potential bias of the classifier. On all datasets we find that GraphEx is more computationally efficient, can generate more diverse explanation graphs and achieves competitive accuracy with other state of the art methods. In Fig1 we show explanations which have been classified with 100% accuracy to their target class. We give the quantitative results of our method and comparison with XGNN in Appendix C

ACKNOWLEDGEMENTS

SB acknowledges the JC Bose Fellowship grant No. JBR/2021/000036/SSC from SERB, GoI.

URM STATEMENT

The authors acknowledge that at least one key author of this work meets the URM criteria of ICLR 2023 Tiny Papers Track.

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

## A  LEARNING THE PARAMETERS OF THE GENERATIVE MODEL

We consider each node $x_i$ and edge $e_{ij}$ in the graph representation model as a random variable that follows a Categorical distribution. Formally,

$$\begin{cases} x_i \sim Categorical(\boldsymbol{\theta}_i) & \boldsymbol{\theta}_i = \{\theta_{i0}, ......, \theta_{ik}\} \\ e_{ij} \sim Categorical(\boldsymbol{\eta}_{ij}) & \boldsymbol{\eta}_{ij} = \{\eta_{ij0}, ....., \eta_{ije}\} \end{cases} \tag{3}$$

We use $I$ to denote the indicator function. Assuming, the above parametric form of the model, the probability of node $x_i$ of belonging to type $a$ can be summarised using the following equation.

$$P(x_i = a) = \prod_{j=0}^{k} \theta_{ij}^{I(a=j)} \tag{4}$$

Similarly, the probability of an edge $e_{ij}$ of belonging to type $a$ can be written as:

$$P(e_{ij} = a) = \prod_{k=0}^{e} \eta_{ijk}^{I(a=k)} \tag{5}$$

Let $D$ denote a collection of graphs with atmost $N$ nodes. A graph $G$ from this collection can be represented using a tuple $(X, E)$ in accordance with the graph representation model described in the paper. Let us denote the set of parameters of the graph representation model using $(\theta, \eta)$. $x_a^{[i]}$ denotes the representation of the $i^{th}$ node in the $a^{th}$ graph of the dataset according to the representation model. Similarly, $e_a^{[ij]}$ denotes the edge between node $i$ and node $j$ of the $a^{th}$ graph of the dataset in the graph representation. Then, the likelihood function can be written as:

$$L(D; \theta, \eta) = \prod_{a=1}^{|D|} \prod_{i=1}^{N} \prod_{j=0}^{k} \theta_{ij}^{I(x_a^{[i]}=j)} \prod_{i=1}^{N} \prod_{j=1}^{N} \prod_{l=0}^{e} \eta_{ijl}^{I(e_a^{[ij]}=l)}$$
$$= \prod_{a=1}^{|D|} \prod_{i=1}^{N} \prod_{j=0}^{k} \theta_{ij}^{I(x_a^{[i]}=j)} \prod_{a=1}^{|D|} \prod_{i=1}^{N} \prod_{j=1}^{N} \prod_{l=0}^{e} \eta_{ijl}^{I(e_a^{[ij]}=l)} \tag{6}$$

Since, we typically work with log-likelihood when finding the maximum likelihood estimate, the log-likelihood function can be written as:

$$l(D; \theta, \eta) = \log L(D; \theta, \eta) = \sum_{a=1}^{|D|} \sum_{i=1}^{N} \sum_{j=0}^{k} I(x_a^{[i]} = j) \log \theta_{ij} \sum_{a=1}^{|D|} \sum_{i=1}^{N} \sum_{j=1}^{N} \sum_{l=0}^{e} I(e_a^{[ij]} = l) \log \eta_{ijl} \tag{7}$$

## A.1 Maximum Likelihood Estimate

We want to find out the parameters($\theta$,$\eta$) of the graph representation model under which the given dataset is most "likely". Formally, we want to find out parameters ($\theta^*$, $\eta^*$) such that:

$$(\theta^*, \eta^*) = \arg\max_{\theta, \eta}(l(D, \theta, \eta)) \tag{8}$$

## A.2 Constraints

However, the log-likelihood function above has to be maximized subject to some constraints as the $\theta$ and $\eta$ parameters belong to a categorical distribution. Since, the parameters of a categorical distribution should sum upto 1, the constraints are as follows:-

$$\begin{cases} g(\boldsymbol{\theta}_i) = \sum_{j=0}^{k} \theta_{ij} - 1 = 0 & 1 \leq i \leq N \\ h(\boldsymbol{\eta}_{ij}) = \sum_{l=0}^{e} \eta_{ijl} - 1 = 0 & 1 \leq i \leq N, 1 \leq j \leq N \end{cases} \tag{9}$$

## A.3 Augmented Target for Unconstrained Optimization

We use an augmented target $\hat{l}(D; \theta, \eta)$ to formulate the likelihood maximization as an unconstrained optimization problem using Lagrange multipliers.

$$\hat{l}(D; \theta, \eta) = l(D; \theta, \eta) + \sum_{i=1}^{N} \lambda_i g(\boldsymbol{\theta}_i) + \sum_{i,j=1}^{N} \beta_{ij} h(\boldsymbol{\eta}_{ij}) \tag{10}$$

We take the derivative and set it to 0 to find out the maxima of each parameter.

$$\begin{cases} \frac{\partial \hat{l}}{\partial \theta_{ij}} = \sum_{a=1}^{|D|}(I(x_a^{[i]} = j)\frac{1}{\theta_{ij}}) - \lambda_i = 0 \\[2mm] \frac{\partial \hat{l}}{\partial \eta_{ijl}} = \sum_{a=1}^{|D|}(I(e_a^{[ij]} = j)\frac{1}{\eta_{ijl}}) - \beta_{ij} = 0 \\[2mm] \frac{\partial \hat{l}}{\partial \lambda_i} = \lambda_i - \sum_{j=0}^{k} \theta_{ij} \\[2mm] \frac{\partial \hat{l}}{\partial \beta_{ij}} = \beta_{ij} - \sum_{l=0}^{e} \eta_{ijl} \end{cases} \tag{11}$$

## A.4 Trick to get rid of the Lagrange Multipliers

We show how to get rid of the Lagrange multipliers to obtain the value of the parameters. We demonstrate the trick to eliminate a $\lambda_i$ to obtain the value of a $\theta_{ij}$, the exact same trick applies to eliminating a $\beta_{ij}$ to obtain the value of a $\eta_{ijl}$. From equation11 we see,

$$\lambda_i = \frac{\sum_{a=1}^{|D|}(I(x_a^{[i]} = j))}{\theta_{ij}} \tag{12}$$

We denote by $N_{ij}$, the numerator of the above equation. Hence,

$$\lambda_i = \frac{N_{ij}}{\theta_{ij}} \tag{13}$$

Notice that, the above equation is valid for all $j \in \{0, 1, ..., k\}$. Hence,

$$\lambda_i = \lambda_i.1 = \lambda_i \sum_{j=0}^{k} \theta_{ij} = \sum_{j=0}^{k} \lambda_i \theta_{ij} = \sum_{j=0}^{k} \frac{N_{ij}}{\theta_{ij}} \theta_{ij} = \sum_{j=0}^{k} N_{ij} = |D| \tag{14}$$

Hence,

$$|D| = \frac{N_{ij}}{\theta_{ij}} \tag{15}$$

$$\theta_{ij} = \frac{N_{ij}}{|D|} \tag{16}$$

Hence, the final MLE estimates are:

$$\begin{cases} \theta_{ij} = \sum_{a=1}^{|D|} \frac{I(x_a^{[i]}=j)}{|D|} \\ \eta_{ijl} = \sum_{a=1}^{|D|} \frac{I(e_a^{[ij]}=l)}{|D|} \end{cases} \tag{17}$$

## B  EXPERIMENTAL DETAILS

### B.1  SYNTHETIC DATA

We use the Networkx[1] library to generate the **Wheel-Tree** dataset. Each graph in the dataset can have between 3 to 10 nodes decided by a random number generator. We generate 100 such graphs for both classes. We shuffle the dataset and have 150 graphs in the training set and 50 graphs in the test set. As the classifier, use a 3 layer graph convolutional network(**GCN**)(Kipf & Welling, 2016) each with a hidden dimension of 64 with ReLU non-linearity after each layer followed by a linear layer which outputs the class logits.

For the **ColoredCycles** dataset we generate the cycle graphs using the Networkx library and add a categorical edge color attribute. Cycles with edge attribute 0 are red cycles and cycles with edge attribute 1 are green cycles. We generate 100 cycles each containing a maximum of 10 nodes for each class. Each graph in the dataset can have between 3 to 10 nodes decided by a random number generator. The GNN classifier for this task consists of a **NNConv**(Gilmer et al., 2017) layer of width 64, a global mean pooling layer followed by a linear layer which outputs the class logits. We use the Adam optimizer with a learning rate varying from 0.01 to 0.0001 in discrete steps with an interval of 10 epochs for both datasets .

### B.2  MUTAG

We use the dataset from the pytorch-geometric repository of graph datasets[2]. The GNN graph classifier for this task consists of 3 consecutive 64 dimensional **GCN** layers, a global mean pooling layer, followed by a linear layer that gives the class logits. We use the Adam Optimizer with a learning rate of 0.01 to train the classifier.

Table 1: Dataset properties and Classifier accuracy

| Dataset | NumClasses | Node Features | Edge Features | GNN Classifier Type | Test Accuracy |
|---|---|---|---|---|---|
| Wheel-Tree | 2 | No | None | GCN | 0.88 |
| ColoredCycles | 2 | No | Red/Green | NNConv | 1.00 |
| MUTAG | 2 | Yes | No | GCN | 0.84 |

## C  COMPARISON WITH STATE OF THE ART

We compare our method on different fronts with the current state of the art model level explanation method XGNN. We have several methodological advantages over XGNN such as time efficiency, one shot generation of graphs, increased diversity of generated explanations generated for a particular class. Our method can generate graphs with different edge attributes which XGNN cannot. Also, XGNN uses another black box deep model to explain the GNN classifier which our method does not.

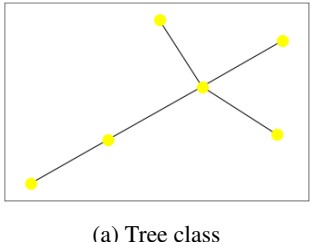
(a) Tree class

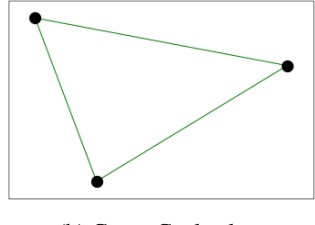
(b) Green Cycle class

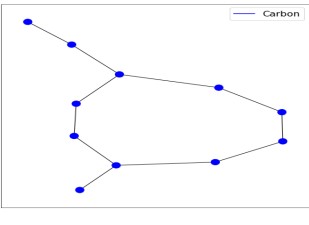
(c) Nonmutagenic class

Figure 2: Explanations for different target classes

Table 2: Results on Wheel-Tree dataset

| | Wheel Class | | | Tree Class | | |
|---|---|---|---|---|---|---|
| Method | Accuracy | Diversity | Training Time | Accuracy | Diversity | Training Time |
| Ours | $0.95 \pm 0.03$ | 0.723 | 4s | $0.97 \pm 0.023$ | 0.863 | 9s |
| XGNN | $0.32 \pm 0.06$ | 0.287 | 29s | $0.91 \pm 0.042$ | 0.349 | 22s |

Table 3: Results on MUTAG dataset

| | Mutagenic Class | | | Non-Mutagenic Class | | |
|---|---|---|---|---|---|---|
| Method | Accuracy | Diversity | Training Time | Accuracy | Diversity | Training Time |
| Ours | $0.75 \pm 0.03$ | 0.756 | 7s | $0.94 \pm 0.016$ | 0.863 | 9s |
| XGNN | $0.91 \pm 0.06$ | 0.213 | 34s | $0.95 \pm 0.043$ | 0.297 | 31s |

Table 4: Results on ColoredCycles dataset

| | Red Cycle Class | | | Green Cycle Class | | |
|---|---|---|---|---|---|---|
| Method | Accuracy | Diversity | Training Time | Accuracy | Diversity | Training Time |
| Ours | $0.1 \pm 0.00$ | 0.526 | 3s | $0.1 \pm 0.00$ | 0.568 | 5s |

We compare our method against XGNN on the **Wheel-Tree** dataset as shown in Table 2 and the **MUTAG** as shown in Table 3 dataset on different metrics. Note, that comparison on **ColoredCycles** dataset is not possible as XGNN cannot generate graphs with edge features. Fig2 shows explanation graphs which were classified to their respective target class with 100% accuracy.

$$Diversity = \frac{D_c}{N_c} \tag{18}$$

We define diversity as the ratio between the number of distinct graphs generated by the explainer for a target class $D_c$ and the total number of explanation graphs produced for that class $N_c$. Generating diverse explanations is crucial since different examples can offer new insights into what the model has learnt about a target class. Also, if a common pattern appears across distinct examples, it suggests that the model's predWeictions for that class heavily rely on that pattern. Our method outperforms XGNN on all metrics on the **Wheel-Tree** dataset. In particular, XGNN struggles to produce class consistent examples for the **Wheel** class. On the MUTAG dataset, XGNN perfornms better on the accuracy metric than our method for the mutagenic class. However, focusing solely on accuracy is misleading as XGNN has very low diversity in the graphs it generates. In other words, it produces few good explanation graphs repeatedly, hence, it does better on the accuracy metric than our method.

---

[1]https://networkx.org/

[2]https://pytorch-geometric.readthedocs.io/en/latest/modules/datasets.html

## D    DIFFERENCES BETWEEN GRAPHEX AND GNN INTERPRETER

GraphEx employs a likelihood maximization scheme to generate an explanation graph for the target class, whereas GNN Interpreter utilizes a numerical optimization scheme for generating explanation graphs. Unlike GNN Interpreter, GraphEx does not require access to the underlying node embeddings when generating graphs for the target class. However, GNN Interpreter has the capability to generate graphs with continuous node feature values, whereas GraphEx, in its current state, can only produce graphs with discrete node features.

