# OpenReview forum: "GraphEx: A User-Centric Model-Level Explainer for Graph Neural Networks"
_ICLR.cc/2023/TinyPapers — Submitted to Tiny Papers @ ICLR 2023_

### Official Review · Reviewer_Qo4F · 2023-03-29

**Confidence:** 3

**Summary Of Contributions:**

This paper proposes a model-level explainer for graph neural network. It learns a graph generative model (GraphEx) to approximate the distribution of graphs classified into a target class by the GNN model. Then, GraphEx can construct representative examples belonging to the target class or the decision boundary.

**Rating:**

Clear, Correct, and Reproducible (CCR): a submission which meets the reviewing criteria

**Strengths And Weaknesses:**

[Strength]:
- The idea of providing insight into the model decision-making process rather than providing explanations for specific input data is promising
- It does not require another black box deep model to explain the GNN and can generate a diverse set of explanation graphs with different node and edge features in one shot.

[Weakness]:
- The proposed method seems to be similar to the GNNInterpreter, but the differences between them have not been fully discussed.
- The experimental results given in the paper are insufficient, so the effectiveness of the methods proposed in the paper cannot be determined.


**Suggested Changes:**

- In fact, the proposed method may have some limitations on the applicable dataset, such as whether a graph is a Gilbert random graph and the type of features of nodes and edges (continuous or concrete). I hope to see more relevant discussions.
- In the experimental part, it only compared with XGNN, which has been proposed for nearly three years. However, there are many updated and effective model-level interpretation methods, such as GANExplainer, Gem and GNNInterpreter, which are also based on generating models. I recommend to supplement experiments to compare with them.

---

### Official Review · Reviewer_U2Fr · 2023-03-29

**Confidence:** 3

**Summary Of Contributions:**

The authors present GraphEx, a model-level explainer for Graph Neural Networks (GNNs) that learns a graph generative model to approximate the distribution of graphs classified into a target class. GraphEx stands out by not requiring a black box deep model or white box access to the GNN, and it can generate diverse explanation graphs in one shot.

**Rating:**

High Potential (HP): a submission which meets the reviewing criteria and has potential to make an impact on the field

**Strengths And Weaknesses:**

Strengths:

1. GraphEx's unique approach eliminates the need for an additional black box deep model to explain the GNN.

2. The ability to generate diverse explanation graphs with varying node and edge features in a single shot enhances the explainer's efficiency.

3. GraphEx does not require white box access to the GNN model, making it more user-friendly and applicable to a wider range of end-users.

**Suggested Changes:**

None

---

### Meta-Review · Area_Chair_5shn · 2023-04-04

**Recommendation:** Invite to present
**Confidence:** 4

**Metareview:**

Good paper with all reviewers arguing for acceptance. Discussion on the difference between GraphEx and GNNInterpreter is suggested.

**Summary:**

the authors proposed a model-level explainer that learns a graph generative model to approximate the distribution of graphs classified into a target class by the GNN model.

**Reason For Not Giving A Higher Recommendation:**

More discussion is needed.

**Reason For Not Giving A Lower Recommendation:**

Clear results.

---

> ### Author Response · Authors · 2023-05-18
> **Reply to Area Chair 5shn**
>
> We thank the reviewers and the meta reviewer for their valuable comments and suggestions.  As suggested,  we have incorporated a discussion on the differences between GraphEx and GNN Interpreter in the appendix.  We could not incorporate this in the body of the paper due to space constraints. We are thankful for the opportunity to present our work at this prestigious venue.

---

### Decision · Program_Chairs · 2023-04-08

Invite to present

---

> ### Author Response · Authors · 2023-05-30
> **Archivization of this paper**
>
> We wish to opt in for archival of our paper